# Low-Dose Aspirin Use Significantly Improves the Survival of Late-Stage NPC: A Propensity Score-Matched Cohort Study in Taiwan

**DOI:** 10.3390/cancers12061551

**Published:** 2020-06-12

**Authors:** Sheng-Dean Luo, Wei-Chih Chen, Ching-Nung Wu, Yao-Hsu Yang, Shau-Hsuan Li, Fu-Min Fang, Tai-Lin Huang, Yu-Ming Wang, Tai-Jan Chiu, Shao-Chun Wu

**Affiliations:** 1Department of Otolaryngology, Kaohsiung Chang Gung Memorial Hospital and Chang Gung University College of Medicine, Kaohsiung 833, Taiwan; rsd0323@cgmh.org.tw (S.-D.L.); jarva@adm.cgmh.org.tw (W.-C.C.); taytay@cgmh.org.tw (C.-N.W.); 2Graduate Institute of Clinical Medical Sciences, College of Medicine, Chang Gung University, Taoyuan 333, Taiwan; 3Department of Traditional Chinese Medicine, Chang Gung Memorial Hospital, Chiayi 613, Taiwan; gmailr95841012@adm.cgmh.org.tw; 4Health Information and Epidemiology Laboratory of Chang Gung Memorial Hospital, Chiayi 613, Taiwan; 5School of Traditional Chinese Medicine, College of Medicine, Chang Gung University, Taoyuan 333, Taiwan; 6Department of Hematology-Oncology, Kaohsiung Chang Gung Memorial Hospital and Chang Gung University College of Medicine, Kaohsiung 833, Taiwan; lee0624@cgmh.org.tw (S.-H.L.); victor99@cgmh.org.tw (T.-L.H.); 7Department of Radiation Oncology, Kaohsiung Chang Gung Memorial Hospital and Chang Gung University College of Medicine, Kaohsiung 833, Taiwan; fang2569@cgmh.org.tw (F.-M.F.); scorpion@cgmh.org.tw (Y.-M.W.); 8Department of Anesthesiology, Kaohsiung Chang Gung Memorial Hospital and Chang Gung University College of Medicine, Kaohsiung 833, Taiwan

**Keywords:** nasopharyngeal carcinoma, low-dose aspirin, overall survival rate, disease-specific survival rate, 10-year follow-up time

## Abstract

*Background*: Aspirin use has been associated with improved survival rates in various cancers. However, it remains unclear if aspirin confers a survival benefit on patients with nasopharyngeal carcinoma (NPC). The aim of this study was to assess the associations between aspirin use and survival in different stages of NPC. *Methods*: This is a 10-year retrospective cohort study of NPC patients. A total of 565 NPC patients were recruited after we performed a 1:4 propensity score match between aspirin users and non–users. Cox regression models with adjusted covariates were employed to evaluate factors that influence the survival rate of NPC patients. *Results*: The Kaplan-Meier analysis revealed that the overall survival (*p* < 0.0001) and disease-specific survival (*p* < 0.0001) rates of 180-day aspirin users increased. Increased survival rates were also observed in 180-day aspirin users with Stages III and IV, T, N1 and 2, and N3 categories. Cox regression models indicated that factors, including aspirin use (univariate: HR = 0.28, 95% CI = 0.14–0.55, *p* < 0.001; multivariate: HR = 0.23, 95% CI = 0.12–0.46, *p* < 0.001), were independent prognostic factors for survival. *Conclusions*: Aspirin use for more than 180 days is associated with an increased survival rate and is a positive independent prognostic factor in NPC.

## 1. Introduction

Nasopharyngeal carcinoma (NPC) is a malignant tumor in the head and neck areas. Unlike other head and neck cancers, NPC is characterized by increased invasiveness and metastasis [1,2,3,4]. Clinical presentation of NPC is highly varied and often results in delayed or missed diagnosis due to its anatomical position [5,6]. NPC is largely undetected and diagnosis is delayed until the tumor reaches an advanced stage, at which time distal-organ metastasis is common. It is estimated that 30–60% of patients diagnosed with NPC will develop metastatic cancers, which account for the vast majority of NPC deaths [2,6,7].

Although NPC is rare in most parts of the world, it has a higher regional incidence in Asia than in Western countries [2,8,9]. The incidence rates of NPC in Southern China and Southeast Asia are among the highest in the world [2,8,9]. Approximately 25–50 cases per 100,000 individuals are reported each year in these regions, while Europe and North America have the lowest incidence rates of fewer than one case per 100,000 individuals each year. In Taiwan, the incidence rate is about 5–10 cases per 100,000 individuals each year, which is considered to be an intermediate risk for NPC [2,8,9]. Considering NPC prevalence is classified as intermediate, and treatment advancement trials for NPC are relatively low compared to other cancers that are common in Taiwan, such as colorectal, lung, and liver cancers [10], we sought to examine the association between aspirin use and the survival rates of patients with NPC in our healthcare system in order to improve the prognosis and outcome for patients with late-stage NPC [8,11,12].

Despite the fact that the five-year survival rate for people diagnosed with early-stage NPC is 72%, the survival rate is dramatically lower in patients with late-stage NPC due to its anatomical position as it relates to detection and management [2,13]. Furthermore, radiotherapy (RT) or concurrent chemoradiotherapy (CCRT) are the standard treatments for NPC that are effective in helping patients with early-stage NPC control their disease [6]. However, the effectiveness of these therapies in treating patients with late-stage NPC is very limited [2,6,7]. Consequently, treatments that can improve NPC survival rates are highly desirable.

Aspirin is widely used as a drug to prevent cardiovascular diseases and has been shown to possess anti-inflammatory properties [14]. The anti-inflammatory properties of aspirin exert favorable effects on managing the challenges of survivorship in various cancers [15,16,17]. A prolongation of survival among aspirin users was seen across different tumor types, such as colorectal, gastrointestinal, and other cancers [18,19,20,21,22,23,24,25,26]. In addition, several retrospective studies demonstrated that cancer patients who have taken aspirin might be at a lower risk of developing metastatic cancer [21,27]. Nevertheless, despite the growing number of randomized controlled trials and meta-analyses of these studies that demonstrate the benefits of aspirin in various cancers [18,19,20,21,22,23,24,25,26,27], the effect of aspirin on NPC has not been explored.

Our interest in evaluating the potential of aspirin to improve the clinical outcomes of NPC patients increased owing to the fact that it improves the survival and cancer management in patients with other cancers [18,19,20,21,22,23,24,25,26,27]. In order to understand whether there is an association between increased survival rates in NPC and aspirin use, we used a database from Chang Gung Memorial Hospital to conduct a cohort study with a long follow-up period (10 years) to analyze overall survival (OS) and disease-specific survival (DSS) rates [28].

## 2. Results

### 2.1. Demographic and Clinical Characteristics of the Study Cohort

The cohort was comprised of a total of 2666 patients who were diagnosed with NPC in Chang Gung Memorial Hospital, Taiwan, between January 2007 and December 2017. Figure 1 shows a flow chart of the cohort study design for statistical analysis. Among the 2666 NPC patients, a total of 463 patients were not recruited after applying the exclusion criteria. We further performed propensity score matching (Appendix A) to avoid an imbalanced covariate distribution between aspirin users and non-users: A total of 565 patients were recruited for this cohort study, of which 113 were identified as aspirin users, and 452 were non-users after their diagnosis of cancer.

Demographics and clinical characteristics from these 565 NPC patients, such as age, sex, AJCC stages of cancer, treatments, lifestyle risk factors, and comorbidities, are summarized in Table 1. In brief, the age, sex, AJCC stages of cancer, and lifestyle risk factors did not differ between the aspirin users and non-users (Table 1). There were significantly higher numbers of aspirin users with comorbidities, such as cerebrovascular accident (CVA), diabetes mellitus (DM), hypertension, and hyperlipidemia, compared to non-users. However, the proportion of NPC patients with atrial fibrillation (i.e., atrial flutter) was similar between aspirin users and non-users. Treatments included CCRT, RT, chemotherapy (CT), and non-treatment, and were significantly different between the two groups (i.e., by comparing the proportions, which include CCRT, RT, CT and none treatments, between the two groups) (Table 1).

### 2.2. Univariate and Multivariate Analyses of Independent Prognostic Factors for Survival

Regarding the influence of prognostic factors on survival rates, the univariate Cox regression analysis showed that various clinical variables, including age (HR_≤60 vs. >60_ =2.05, 95% CI = 1.43–2.95, *p* < 0.001), AJCC stages of cancer (HR_stages I and II vs. stages III and IV_ = 3.88, 95% CI = 2.45–6.15, *p* < 0.001), CT (HR _CT treatment vs. CCRT standard NPC treatment_ = 29.63, 95% CI = 12.14–72.34, *p* < 0.001), and aspirin use (HR_aspirin non-users vs. users_ = 0.28, 95% CI = 0.14–0.55, *p* < 0.001) were significantly associated with the survival rate, while patient sex (HR_females vs. males_ = 1.26, 95% CI = 0.76–2.11, *p* = 0.372) and RT (HR_RT treatment vs. CCRT standard NPC treatment_ = 0.78, 95% CI = 0.44–1.39, *p* = 0.405) were not significantly related to survival. The multivariate Cox regression analysis, the results of which were similar to those of the univariate analysis, indicated that the factors of age (HR_≤60 vs. >60_ = 2.11, 95% CI = 1.45–3.09, *p* < 0.001), AJCC stages of cancer (HR_stages I and II vs. stages III and IV_ = 4.02, 95% CI = 2.43–6.67, *p* < 0.001), CT (HR_CT treatment vs. CCRT standard NPC treatment_ = 25.34, 95% CI = 9.83–65.32, *p* < 0.001) and aspirin use (HR_aspirin non-users vs. users_ = 0.23, 95% CI = 0.12–0.46, *p* < 0.001) were still significantly associated with survival. In contrast, patient sex (HR = 1.08_females vs. males_, 95% CI = 0.64–1.82, *p* = 0.773) and RT (HR_RT treatment vs. CCRT standard NPC treatment_ = 1.29, 95% CI = 0.67–2.48, *p* = 0.454) were not independent prognostic factors for survival (Table 2).

Comorbidities were also additionally adjusted in univariate and multivariate Cox regression analyses after considering aspirin use in the 113 patients may be attributable to the comorbidities related to cardiovascular diseases in our study (Appendix A). Age, AJCC stages of cancer, CT treatment, and aspirin use were still significantly associated with survival regardless of adjustment with (Appendix A) or without comorbidities (Table 2). In addition, we further evaluated the influence of each of the comorbidity on the association between aspirin use and NPC cancer survival (Appendix A). The HR of aspirin use was calculated by taking either each of the comorbidity or all of the comorbidities into account (Appendix A). After the adjustments, our results showed that the HRs of aspirin use varied from 0.19 to 0.40 with *p*-values lesser or equal to 0.022, indicating that the comorbidities tended to slightly alter the associations between aspirin use and NPC cancer survival. Overall, our results demonstrated that aspirin use is still an independent prognostic factor for NPC survival.

### 2.3. Survival Analyses

In this cohort of patients, there were a total of 184 deaths (32.6%) in 565 patients. Regarding aspirin use, there were 17 (15%) deaths in the 113 patients who were aspirin users and 184 (32.6%) deaths in the 448 patients who were non-users (Table 1). We performed a Kaplan–Meier survival analysis to further evaluate whether low-dose aspirin use (100 mg/day) for more than 180 days is associated with an increased OS rate in patients of all ages with NPC after a diagnosis of cancer. We performed a Kaplan–Meier survival analysis to further evaluate whether low-dose aspirin use (100 mg/day) for more than 180 days is associated with an increased OS rate in patients of all ages with NPC after a diagnosis of cancer. The Kaplan–Meier survival analysis demonstrated that patients with NPC who received aspirin for at least 180 days had better survival rates compared to non-users (Figure 2). Similarly, aspirin users also had significantly better DSS rate when we applied Kaplan–Meier survival curves to analyze the NPC-specific survival rate in this cohort study (Figure 3). Interestingly, in contrast to patients with NPC who received aspirin for at least 180 days, patients who only received aspirin for either 30–89 days or 90–179 days had no significantly higher DSS rate compared to non-users (Appendix A).

Survival benefits associated with at least 180 days of aspirin use were observed in patients with Stages III and IV NPC (*p* < 0.001), while no statistical significance was observed in patients with Stages I and II NPC (*p* = 0.302) (Table 3). We also evaluated the effect of aspirin on the survival rates of NPC patients in the T and N categories. Aspirin treatment outcomes for NPC patients in either T (T1 and 2 and T3 and 4) (Table 4) or N (N1 and 2 and N3) category (Table 5) in terms of survival rates were significantly improved. Overall, these stratifiable analyses also suggested that patients with late-stage NPC were the most likely to benefit from aspirin use when patients received aspirin for more than 180 days after NPC diagnosis.

## 3. Discussion

Aspirin is relatively safe and affordable and can help ease the burden of long-term medical care of diseases. It has shown promising beneficial effects for treatments across a broad range of diseases [15,16,17,18,19,20,21,22,23,24,25,26,27,29,30]. Indeed, aspirin use has been shown to make significant advances in reducing the risk of developing cancers and cancer-related deaths over the past few years. [15,16,17,18,19,20,21,22,23,24,25,26,27]. Its valuable role as a treatment for cancer has been widely documented in many cancer-related studies [15,16,17,18,19,20,21,22,23,24,25,26,27]. Despite studies that specifically demonstrate the adverse effects of aspirin [14,15,16,17,18,19,20,21,22,23,24,25,26], there has been increasing interest in the role of aspirin in various cancer treatments [15,16,17,18,19,20,21,22,23,24,25].

Due to detection and management limitations, late-stage NPC is considered to be a serious health problem in Taiwan and is the leading cause of cancer-related death [2,6,7,8,9,10,11,12,13]. It is not known whether use of low-dose aspirin after the diagnosis of NPC is associated with OS and DSS rates in patients with NPC. To address this gap, therefore, we conducted this study by collecting data from the Chang-Gung healthcare system, which provided comprehensive data of real-world patients in a clinical practice. The primary advantages of using this database are that we were ensured high follow-up rates of patients for a longer period of time (10 years), and because of the data-quality needed for this retrospective study [28], we had the ability to control potential selection bias with the following covariates: age, AJCC stages of cancer and treatments. Hence, in this retrospective analysis, we followed-up with patients with NPC at different stages after their diagnosis for a 10-year period, instead of only five years, and in sufficient numbers to gather accurate statistics on long-term OS and DSS rates.

In this study, we attempted to minimize the bias between aspirin users and non-users by using a propensity score match at a 1:4 ratio. As one of the results from this matching, the baseline patient characteristics revealed that the patients who had aspirin prescribed were more likely to have comorbidities related to cardiovascular diseases, such as CVA, DM, hypertension, and hyperlipidemia, compared to non-users (Table 1). This result was unsurprising and was expected due to our study design, given that aspirin is known to have beneficial effects in persons with cardiovascular diseases; therefore, those who were identified as aspirin users in our cohort study were usually thus because of prevalent comorbidities related to cardiovascular diseases. Conversely, those recruited NPC patients who were non-users did not have cardiovascular-related comorbidities, and thus might have been less likely to take aspirin after their cancer diagnosis. This finding is consistent with other studies that have demonstrated that aspirin users who are recruited for retrospective studies display significantly more comorbidities related to cardiovascular diseases at their baseline [25,26,29], which suggests that our adopted approach of using a propensity score match at a 1:4 ratio for this study was reliable and reflective of clinical scenarios. In addition, our results suggested that aspirin use is a good prognostic factor for NPC survival after carefully considering and adjusting for the potential sources of bias and confounding effects in different models (Table 2, Appendix A).

We observed that aspirin use for at least 180 days after diagnosis of NPC positively affected 10-year OS and DSS rates (Figure 2, Figure 3, Appendix A); this effect was most pronounced for patients with late-stage NPC (Table 3). This was also true for subgroups of patients defined by a TN staging system; our data showed that aspirin use after diagnosis was associated with increased survival rates among NPC patients with either T (T1 and 2 and T3 and 4) (Table 4) or N (N1 and 2 and N3) category (Table 5). Since the T category is associated with local containment of tumor, and the N category is associated with distant metastasis, our data suggested that aspirin use showed beneficial effects for patients related to either local containment of the tumor or distant metastasis. In addition, while this database contained information for the duration of aspirin treatment, we were also able to test whether there was any benefit of taking aspirin for 30–89 days, 90–179 days (Appendix A), or more than 180 days (Figure 2, Figure 3, Appendix A). Our analyses suggested that there is only an improvement in the 10-year survival rates (OS and DSS) for aspirin users who received aspirin more than 180 days. All of these findings shed light on whether or not aspirin can be used as an additional cancer treatment, especially in patients with late-stage NPC after careful consideration of risks related to side effects of aspirin, such as hemorrhaging [15].

It has been suggested that age plays a role in cancer survival [31,32,33,34,35]. However, the influence of age on cancer survival outcome is still considered controversial [31,32,33,34,35]. There are some studies that suggest that a younger age at diagnosis has been demonstrated to be a significant poor prognostic factor for cancer survival in a number of cancers [34,35], while other studies do not show this [31,32,33]. With respect to NPC, our results were consistent with prior studies, in which age had a direct effect on NPC survival rates [31,32,33]. Patient age showed statistical significance in both univariate and multivariate analyses in our study (Table 2). Considering that elderly patients are more likely to have functional decline and comorbidities, it is great to have less aggressive therapies or abbreviated treatment courses, which at the same time still generate overall benefits and improve cancer management and survivorship in elderly patients with NPC. Our findings identified factors, such as aspirin use, that are associated with improved cancer survival rate, suggesting that this information can be used for future improvements in NPC care, particularly for elderly patients with late-stage NPC.

Several studies have shown that aspirin use improves survival rates in the late-stage of various types of cancers [15,24,26,27]. However, the precise mechanism that underlies how aspirin affects carcinogenesis and leads to clinically relevant improvements in survival rates remains elusive [14]. Some mechanisms have been proposed as they relate to aspirin use to reduce cancer mortality [14]. For example, research suggests that aspirin regulates pro-inflammation through COX-2-dependent pathways, which ultimately influences multiple pathways involved in tumor cell proliferation and metastasis [14,36,37]. In fact, it has been suggested that a COX-2 pathway blockade could be a potential anti-tumor treatment in patients with recurrent NPC [37]. In addition, gene mutations, such as *PIK3CA* and *BRAF*, may also have an impact on the benefits of aspirin use for several types of cancer. [38,39]. Interestingly, a study showed higher expression of PIK3CA as being significantly associated with advanced NPC, thereby suggesting that increased expression of PIK3CA may contribute to tumor cell proliferation and metastasis [40]. Collectively, the evidence from these studies may not only explain our findings, but also elicit some questions that are in need of further investigation; for instance, whether aspirin use can benefit NPC patients with high COX-2 or PIK3CA expression. Perhaps, we also could gather more information about the possible mechanisms of aspirin on NPC survival outcomes by exploring databases such as Cancer Genome Atlas (TCGA) and Gene Expression Omnibus (GEO). We anticipate that such studies will contribute to the future development of more effective cancer treatments for patients with NPC.

## 4. Materials and Methods

### 4.1. Patient Recruitment

This cohort study was approved by the Institutional Review Board (IRB) of the Kaohsiung and Chiayi branches of Chang Gung Memorial Hospital with reference numbers 202000714B0 and 201700253B0C602. The requirement for informed consent was waived, according to the nature of the study design and IRB regulations. During the time period of 01/01/2007–12/31/2017, 2666 patients diagnosed with nasopharyngeal malignancy (ICD10: C110, C111, C112, C113, and C119) in the Chang Gung Research Database were identified. Exclusion criteria were AJCC Stage IVc or missing data on stage of cancer (*n* = 290), aspirin use less than 180 days (*n* = 114), nasopharyngeal malignancies with morphology codes other than 8010, 8020, 8070, 8071, 8072, and 8082 (*n* = 38), and less than 20 years of age (*n* = 21). In total, 2203 patients were analyzed in our study.

### 4.2. Statistical Analysis

Categorical data, such as sex, comorbidities, lifestyle risk factors, AJCC stages of cancer, etc., were tested by either a two-sided Fisher’s exact test or a Pearson’s chi-squared test. The normally and non-normally distributed continuous data were analyzed using Student’s t-tests and Mann–Whitney U tests, respectively. In order to minimize the confounding effect of groups that are comparable due to non-randomized allocations, a 1:4 propensity score-matched study group (aspirin user vs. non-user) was created using the Greedy method with a 0.25 caliper-width using NCSS 10 software (NCSS Statistical Software, Kaysville, UT, USA) (Appendix A) [41,42,43,44,45,46,47]. The propensity scores were calculated using a logistic regression model with the following covariates: sex, age and AJCC stages of cancer. After adjusting for these confounding factors, the Kaplan–Meier method was used to evaluate the effects of aspirin use in the primary outcome (DSS). A univariate analysis and Cox proportional-hazards model were used to evaluate any parameters that could affect survival. All statistical analyses were performed using SPSS Statistics V22.0 software for Windows (IBM Corp., Armonk, NY, USA). Statistical significance was set for each analysis at *p*-values of <0.05.

## 5. Conclusions

In summary, we conducted 10-year cohort study using a database from Chang Gung Memorial Hospital, which was an accurate reflection of diverse patient subgroups and representative of the nation’s NPC cancer cases in Taiwan. Our findings indicated that OS and DSS rates exhibited significant improvement in patients with NPC after aspirin use for more than 180 days. According to the univariate and multivariate Cox regression analyses, which confirmed previous findings that factors, such as patient age and AJCC stages of cancer, are significantly associated with survival rates in patients with NPC. Moreover, these analyses contribute additional evidence that suggests aspirin use is an independent prognostic factor for improving survival rates. Taken together, these new understandings will provide the opportunity to improve the management of cancer care and survivorship for patients with late-stage NPC in Taiwan, as well as in the areas where NPC is endemic.

## Figures and Tables

**Figure 1 cancers-12-01551-f001:**
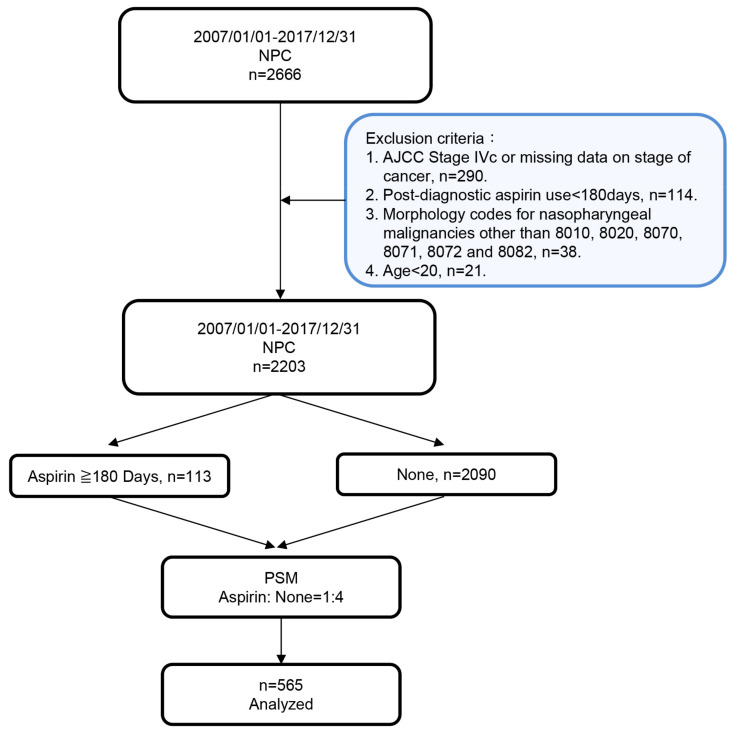
Flowchart of NPC patient inclusion and exclusion in the study cohort from the Chang Gung Memorial Hospital database. A total of 2666 patients diagnosed with NPC were recruited for this study. Aspirin users (≥180 days) were matched with non-users (None) based on a 1:4 propensity score (PSM), resulting in a final inclusion of 565 patients with NPC for data analysis.

**Figure 2 cancers-12-01551-f002:**
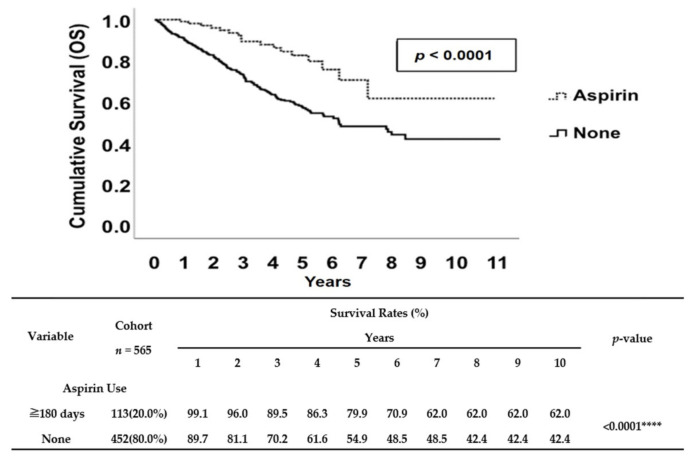
Kaplan–Meier survival curve of OS rates between aspirin users (≥180 days) and non-users. The median OS rates of aspirin non-users (None) was about 5.9 years, and the estimated 5- and 10-year OS rates were 54.9% and 42.4%, respectively. Although median survival was not reached for aspirin users (≥180 days), estimated 5- and 10-year OS rates were 79.9% and 62.0%, respectively. **** indicates *p* ≤ 0.0001.

**Figure 3 cancers-12-01551-f003:**
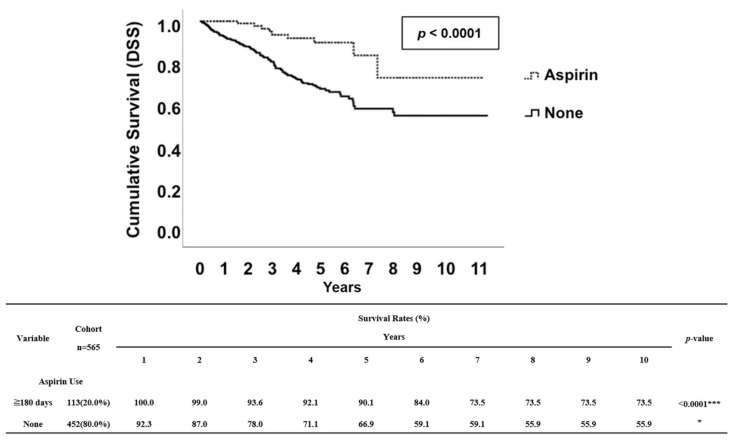
Kaplan–Meier survival curve of DSS rates between aspirin users (≥ 180 days) and non-users. The estimated 5- and 10-year DSS rates of aspirin non-users (None) were 66.9% and 55.9%, respectively. The estimated 5- and 10-year DSS rates of aspirin users (≥180 days) were 90.1% and 73.5%, respectively. **** indicates *p* ≤ 0.0001.

**Table 1 cancers-12-01551-t001:** Demographic and clinical characteristics of the study cohort (*n* = 565).

Variables	Cohort *n* = 565	Low-Dose Aspirin		*p*-Value
		Non-Users *n* = 452	Users ≥ 180 Days *n* = 113	
**Mean age at diagnosis, years** **(IQR)**	51.8 (45.64–60.3)	52.0 (45.3–60.0)	51.0 (47.0–61.5)	0.476
**Sex**				
Female	93 (16.5%)	73 (16.2%)	20 (17.7%)	0.691
Male	472 (83.5%)	379 (83.8%)	93 (82.3%)	
**Age**				
≤60 years	420 (74.3%)	340 (75.2%)	80 (70.8%)	0.335
>60 years	145 (25.7%)	112 (24.8%)	33 (29.2%)	
**Stages of Cancer (AJCC)** ^a^				
I and II	231 (40.9)	183 (40.5%)	48 (42.5%)	0.700
III and IV ^b^	334 (59.1)	269 (59.5%)	65 (57.5%)	
**Cancer Recurrence**				
No	508 (89.9%)	404 (89.4%)	104 (92.0%)	0.402
Yes	57 (10.1%)	48 (10.6%)	9 (8.0%)	
**Mortality**				
Alive	381 (67.4%)	285 (63.1%)	96 (85.0%)	< 0.001 ***
Dead	184 (32.6%)	167 (36.9%)	17 (15.0%)	
**Causes of Death**				
Alive	381 (67.4%)	285 (63.1%)	96 (85.0%)	< 0.001 ***
Dead due to NPC	125 (22.1%)	116 (25.7%)	9 (8.0%)	
Other Causes of Dead	59 (10.4%)	51 (11.3%)	8 (7.1%)	
**Treatments**				
CCRT	470 (83.2%)	365 (80.8%)	105 (92.9%)	0.006 *
RT	80 (14.2%)	73 (16.2%)	7 (6.2%)	
CT	9 (1.5%)	8 (1.8%)	1 (0.9%)	
No	6 (1.1%)	6 (1.3%)	0 (0.0%)	
**Lifestyle Risk Factors**				
Smoking				
No	332 (58.8%)	260 (57.5%)	72 (63.7%)	0.232
Yes	233 (41.2%)	192 (42.5%)	41 (36.3%)	
Betel nuts consumption				
No	455(80.5%)	363(80.3%)	92(81.4%)	0.791
Yes	110(19.5%)	89(19.7%)	21(18.6%)	
Alcoholic beverages				
No	384 (68.0%)	304 (67.3%)	80 (70.8%)	0.471
Yes	181 (32.0%)	148 (32.7%)	33 (29.2%)	
**Comorbidities**				
CVA				
No	451 (79.8%)	406 (89.8%)	45 (39.8%)	< 0.001 ***
Yes	114 (20.2%)	46 (10.2%)	68 (60.2%)	
DM				
No	490 (86.7%)	409 (90.5%)	8 (71.7%)	< 0.001 ***
Yes	75 (13.3%)	43 (9.5%)	32 (28.3%)	
Hypertension				
No	434 (76.8%)	378 (83.6%)	56 (49.6%)	< 0.001 ***
Yes	131 (23.2%)	74 (16.4%)	57 (50.4%)	
Atrial fibrillation (flutter)				
No	559 (98.9%)	448 (99.1%)	111 (98.8%)	0.442
Yes	6 (1.1%)	4 (0.9%)	2 (1.8%)	
Hyperlipidemia				
No	389 (68.8%)	366 (81.0%)	23 (20.4%)	< 0.001 ***
Yes	176 (31.2%)	8 (19.0%)	90 (79.6%)	

Abbreviations: IQR interquartile range; CCRT concurrent chemoradiotherapy; RT radiotherapy; CT chemotherapy; CVA cerebrovascular accident; DM diabetes mellitus; ^a^ AJCC Cancer Staging 7th Edition; ^b^ Stages IVa and IVb only; * *p* ≤ 0.05; *** *p* ≤ 0.001.

**Table 2 cancers-12-01551-t002:** Univariate and multivariate Cox proportional hazards models of prognostic factors for NPC survival.

Variables	Cohort *n* = 565	Hazard Ratio (95%CI)			
		Univariate	*p*-Value	Multivariate	*p*-Value
**Sex**					
Female	93 (16.5%)	1	0.372	1	0.773
Male	472 (83.5%)	1.26 (0.76–2.11)		1.08 (0.640–1.82)	
**Age**					
≤60 years	420 (74.3%)	1	<0.001 ***	1	<0.001 ***
>60 years	145 (25.7%)	2.05 (1.43–2.95)		2.11 (1.45–3.09)	
**Stages of Cancer (AJCC) ^a^**					
I and II	231 (40.9)	1	<0.001 ***	1	<0.001 ***
III and IV ^b^	334 (59.1)	3.88 (2.45–6.15)		4.02 (2.43–6.67)	
**Treatments**					
CCRT	470 (83.2%)	1		1	
RT	80 (14.2%)	0.78 (0.44–1.39)	0.405^c^	1.29 (0.67–2.48)	0.454^e^
CT	9 (1.5%)	29.63 (12.14–72.34)	<0.001 ***^d^	25.34 (9.83–65.32)	<0.001 ***^f^
**Aspirin Use**					
No	452 (80.0%)	1	<0.001 ***	1	<0.001 ***
≥180 days	113 (20.0%)	0.28 (0.14–0.55)		0.23 (0.12–0.46)	

Abbreviations: 95% CI 95% confidence interval; CCRT concurrent chemoradiotherapy; RT radiotherapy; CT chemotherapy; ^a^ AJCC Cancer Staging 7th Edition; ^b^ Stages IVa and IVb only; ^c, e^ Comparing RT to CCRT; ^d, f^ Comparing CT to CCRT; *** *p* ≤ 0.001.

**Table 3 cancers-12-01551-t003:** Comparing mortality rates between aspirin users (≥ 180 days) and non-users in patients (*n* = 565) with Stages I and II and III and IV NPC.

AJCC *n* = 565	Low-Dose Aspirin		X^2^ Aspirin Non-Users vs. Users	*p*-Value	
	Non-Users *n* = 452	Users ≥ 180 Days *n* = 113		Aspirin Non-Users vs. Users	Stages I and II vs. III and IV ^a^
**Stages I and II**					<0.001 ***
Alive	146 (79.8%)	41 (85.4%)	2.395	0.302	
Dead due to NPC	20 (10.9%)	2 (4.2%)			
Other Causes of Dead	17 (9.3%)	5 (10.4%)			
**Stages III and IV^a^**					
Alive	139 (51.7%)	55 (84.6%)	23.364	<0.001 ***	
Dead due to NPC	96 (35.7%)	7 (10.8%)			
Other Causes of Dead	34 (12.6%)	3 (4.6%)			

Abbreviation: vs. versus. ^a^ Stages IVa and IVb only; **** *p* ≤ 0.001.

**Table 4 cancers-12-01551-t004:** Comparing mortality rates between aspirin users (≥180 days) and non-users in patients (*n* = 333) with a subdivision of Stages III and IV NPC in the T category.

Stages III and IV ^a^ *n* = 333	Low-Dose Aspirin		*p*-Value	
	Non-Users ^b^ *n* = 268	Users ≥ 180 Days *n* = 65	Aspirin Non-Users vs. User	T1 and 2 vs. T3 and 4
**T1 and 2**				<0.001 ***
Alive	53 (62.4%)	23 (88.5%)	0.014 *	
Dead due to NPC	25 (29.4%)	3 (11.5%)		
Other Causes of Dead	7 (8.2%)	0 (0.0%)		
**T3 and 4**				
Alive	86 (47.0%)	32 (82.1%)	<0.001 ***	
Dead due to NPC	71 (38.8%)	4 (10.3%)		
Other Causes of Dead	26 (14.2%)	3 (7.7%)		

Abbreviation: vs. versus; ^a^ Stages IVa and IVb only; ^b^ One patient with missing data on T category of cancer; * *p* ≤ 0.05; *** *p* ≤ 0.001.

**Table 5 cancers-12-01551-t005:** Comparing mortality rates between aspirin users (≥180 days) and non-users in patients (*n* = 333) with a subdivision of Stages III and IV NPC in the N category.

Stages III and IV ^a^ *n* = 333	Low-dose Aspirin		*p*-value	
	Non-Users ^b^ *n* = 268	Users ≥ 180 Days *n* = 65	Aspirin Non-Users vs. Users	Comparing N0, N1 and 2, and N3
**N0**				<0.001 ***
Alive	9 (52.9%)	3 (75.0%)	0.293	
Dead due to NPC	5 (29.4%)	0 (0.0%)		
Other Causes of Dead	3 (17.6%)	1 (25.0%)		
**N1 and 2**				
Alive	105 (56.1%)	37 (90.2%)	<0.001 ***	
Dead due to NPC	59 (31.6%)	3 (7.3%)		
Other Causes of Dead	23 (12.3%)	1 (2.4%)		
**N3**				
Alive	25 (39.1%)	15 (75.0%)	0.017*	
Dead due to NPC	32 (50.0%)	4 (20.0%)		
Other Causes of Dead	7 (10.9%)	1 (5.0%)		

Abbreviation: vs. versus; ^a^ Stages IVa and IVb only; ^b^ One patient with missing data on N category of cancer. * *p* ≤ 0.05; *** *p* ≤ 0.001.

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
