# Peer review of "Low-Dose Aspirin Use Significantly Improves the Survival of Late-Stage NPC: A Propensity Score-Matched Cohort Study in Taiwan"

_cancers, 2020, doi:10.3390/cancers12061551_

Round 1

Reviewer 1 Report

  1. The sections are in in standard order. Please revise: introduction, methods and materials, results, discussion.
  2. Line 29. Suggest remove 'especially patients with late-stage NPC' because you make clear at line 79 that NPC survival and aspirin use has not been studied.
  3.  line 36. Why is aspirin use less than 180 days being mentioned here, when you say in the methods section that these patients have been excluded.
  4. line 31. You give the impression here and in other parts of the manuscript that patients have been recruited for this study at time of diagnosis. In fact this is an analysis of historical data, you need to make this clear.
  5. lines 88-99. Most of this is repetition of the methods and materials section, and can be deleted.
  6. line 103. The author note that patients with co-morbidities tend to be aspirin users and that this is what you would expect. But something very wrong is going on in Table 1 where it claims to show that it is unusual to be a CVA patient taking aspirin.
  7. Table 2. It is very helpful to show univariate and multivariate findings in this Table. But it would be very informative to include co-morbidities in this analysis (once the problem in comment 6 is resolved.
  8. It is not necessary to have three places of decimals for the hazard ratios and CIs, two will suffice.

Reviewer 2 Report

Overall comments

Examines the use of aspirin and outcomes of nasopharyngeal carcinoma over 10 years.

It is well designed and written.  Potential sources of bias such as participant selection and confounding should be covered in more detail.

Specific Comments

Abstract needs some numbers in it

Line 111:  3 decimal places not required

Reviewer 3 Report

Luo et al. present a 10-year cohort study to analyse the effect of Aspirin on survival rate of NPC. The authors use collected data from Chang Gung Memorial Hospital Taiwan.

The study is interesting, but needs fundamental improvements:

- The patient subgroups are inbalanced (452 non-users vs. 113 aspirin-users).

- The authors misinterpretated the covariate influence of the CCRT treatment on the cancer survival rate and introduce an optimistic bias of aspirin. There is a significant influence of CCRT on survival rate, and CCRT shows higher proportion in the aspirin-users compared to non-users. Moreover, there is a survival benefit of aspirin-users with stages III+IV compared to stages I+II, whereas stages shows similar proportion in the aspirin-users and the non-users. This indicates that the survival benefit comes from CCRT treatment, but no obvious evidence that aspirin influence the cancer survival rate and functions as an independent prognostic factor for NPC. However, it is more like that aspirin might positively influence/reduce death risc for cardivascular diseases such as CVA, hypertension and hyperlipidemia after aggressive cancer treatment such as CCRT rather then NPC survival rate. This should be further analyzed by the authors, for instance by spliting the data and compare the 5 years with 10 years survival effects. Moreover, the authors should perform a comparison with data such as from TCGA, GEO and cBioPortal to show the potential effect of aspirin (NPC survival rate vs. death risc for cardivascular diseases).

- The study is limited to Taiwan which should be mentioned in the manuscript and title.

Round 2

Reviewer 1 Report

  1. Sections are still not in correct order. The paper should be: introduction, materials and methods, results, discussion, conclusion.
  2. There seems to be a typo in the heading to Table S1, '80' should probably be 180.